# Parallel evolution of *Pseudomonas aeruginosa* phage resistance and virulence loss in response to phage treatment in vivo and in vitro

Meaghan Castledine[1]*, Daniel Padfield[1], Pawel Sierocinski[1], Jesica Soria Pascual[1], Adam Hughes[1], Lotta Mäkinen[1], Ville-Petri Friman[2], Jean-Paul Pirnay[3], Maya Merabishvili[3], Daniel de Vos[3], Angus Buckling[1]

[1]College of Life and Environmental Sciences, Environment and Sustainability Institute, University of Exeter, Penryn, United Kingdom; [2]Department of Biology, University of York, York, United Kingdom; [3]Laboratory for Molecular and Cellular Technology, Queen Astrid Military Hospital, Brussels, Belgium

**\*For correspondence:**
mcastledine96@gmail.com

**Competing interest:** The authors declare that no competing interests exist.

**Abstract** With rising antibiotic resistance, there has been increasing interest in treating pathogenic bacteria with bacteriophages (phage therapy). One limitation of phage therapy is the ease at which bacteria can evolve resistance. Negative effects of resistance may be mitigated when resistance results in reduced bacterial growth and virulence, or when phage coevolves to overcome resistance. Resistance evolution and its consequences are contingent on the bacteria-phage combination and their environmental context, making therapeutic outcomes hard to predict. One solution might be to conduct 'in vitro evolutionary simulations' using bacteria-phage combinations from the therapeutic context. Overall, our aim was to investigate parallels between in vitro experiments and in vivo dynamics in a human participant. Evolutionary dynamics were similar, with high levels of resistance evolving quickly with limited evidence of phage evolution. Resistant bacteria—evolved in vitro and in vivo—had lower virulence. In vivo, this was linked to lower growth rates of resistant isolates, whereas in vitro phage resistant isolates evolved greater biofilm production. Population sequencing suggests resistance resulted from selection on de novo mutations rather than sorting of existing variants. These results highlight the speed at which phage resistance can evolve in vivo, and how in vitro experiments may give useful insights for clinical evolutionary outcomes.

## Editor's evaluation

With the increased interest in phage therapy to treat antibiotic resistant infections, there are questions about the ease at which bacteria evolve phage resistance. To examine this, Castledine et al. cultured a set of bacterial isolates from a patient pre- and during phage therapy and also experimentally evolved a mixture of the bacterial isolates from the patient in the absence or presence of phage in vitro. Overall, the authors observed similarities between the evolutionary outcomes (genomic and phenotypic) in the patient and in vitro.

## Introduction

By 2050, as many as 10 million people may die annually from antimicrobial-resistant infections (*WHO, 2019*). To combat this threat, phage therapy is being increasingly researched as a complement to antibiotics (*Abedon, 2019*; *Kwiatek et al., 2020*). Phage therapy uses viruses (phage) that infect

pathogenic bacteria to clear infections (*Kwiatek et al., 2020*). Phage therapy has several advantages to antibiotics by being species-specific (thus not harming beneficial microbiota) (*Koskella and Meaden, 2013*) and self-amplifying at infection sites (*Górski et al., 2015*). A major consideration with the use of therapeutic phages is the ease and speed (sometimes within hours) at which bacteria can evolve resistance, which can greatly limit the therapeutic potential of phages (*Jariah and Hakim, 2019*; *Torres-Barceló, 2018*; *Torres-Barceló and Hochberg, 2016*). However, phages may contribute to reducing the severity of infections even after resistance evolves. First, phage can sometimes evolve to overcome this resistance (i.e., bacteria-phage coevolution) and so they may continue lowering bacterial densities (*Gandon et al., 2008*; *Hampton et al., 2020*). Second, resistance can trade-off against bacterial growth rates and virulence; as the alteration or loss of bacterial surface structures, typically associated with phage resistance, can affect other fitness-related phenotypes such as nutrient uptake, biofilm formation, and binding of host receptors (*Mangalea and Duerkop, 2020*). Predicting the evolutionary outcome of phage therapy is therefore of crucial importance (*Abedon, 2017*).

Bacteria-phage dynamics and its consequences for phage therapy are highly context-specific (*Gandon et al., 2008*; *Hampton et al., 2020*). Both human clinical and animal models studies report a range of outcomes with differing phage therapy success, and differences in the extent of resistance evolution and associated costs and virulence (*Abedon, 2019*; *Brix et al., 2020*; *Kwiatek et al., 2020*; *Leitner et al., 2021*). One solution to better understand the outcome of in vivo therapy is to conduct controlled in vitro experiments with the relevant organisms. However, while most of our understanding of the ecology and evolution of bacteria-phage interactions comes from highly controlled in vitro environments (*Gandon et al., 2008*; *Hampton et al., 2020*), we do not know how well in vitro dynamics mirror in vivo dynamics quantitatively or even qualitatively.

There are good reasons to expect the outcomes may be very different. In vivo environments are expected to be more stressful for bacteria owing to limiting resources, presence of immune cells, and competing microbiota (*Everest, 2007*). Consequently, these conditions may differ significantly from studies of bacteria-phage coevolution in nutrient-rich microcosms which could drive differences in (co)evolutionary dynamics and associated trade-offs (*Gandon et al., 2008*; *Hampton et al., 2020*). In order to evolve phage resistance, bacteria must have sufficient mutation supply, which will primarily be determined by population size and gene flow (*Gandon et al., 2008*; *Hampton et al., 2020*; *Morgan et al., 2010*; *Morgan et al., 2005*; *Pal et al., 2007*). Smaller population sizes in vivo may therefore limit resistance. Moreover, resistance costs and trade-offs may be greater as a consequence of the additional selection pressures facing bacteria, which may limit selection for resistance, making it easier for phage to overcome resistance, or reduce growth rates and virulence (*Buckling et al., 2006*; *Scanlan et al., 2015*).

Consequently, the main aim of this research was to determine how well in vitro bacteria—phage coevolutionary dynamics mirror dynamics in a clinical context. As a starting point, we followed bacteria-phage coevolutionary dynamics during a phage-mediated decolonization trial in parallel with the dynamics between the same bacteria and phage in vitro. The nasal cavity can act as a source of infection, notably from highly drug-resistant *Pseudomonas aeruginosa* and *Staphylococcus aureus* when patients undergo hospitalization and are treated with broad-spectrum antibiotics (*Mainz et al., 2009*; *Wertheim et al., 2004*). The trial was conducted to try and decolonize intensive care (ICU) patients with *P. aeruginosa* and *S. aureus*, using a phage cocktail prior to antibiotic treatment in the prevention of ICU infections with these pathogens. Mucirocin (trade name Bactroban) and chlorhexidine are routinely used for nasal decolonization in such contexts, but extensive use of these agents has led to resistance and new nasal antiseptics need to be considered (*Edgeworth, 2011*). We focused on a single patient, in which we could detect *P. aeruginosa* at the beginning of the trial and both *P. aeruginosa* and phage during treatment. Decolonization of *P. aeruginosa* was ultimately successful in this patient. We used the pre-treatment *P. aeruginosa* isolates and evolved these populations in vitro in the absence and presence of treatment phages. We measured bacteria-phage (co)evolution and while we found extensive resistance evolution both in vivo and in vitro, we only found evidence of limited phage infectivity evolution in vitro. We determined the consequences of resistance, with respect to virulence, and the possible mechanisms linking this resistance-virulence trade-off using phenotypic (growth rate and biofilm assays) and metagenomic analyses for the in vivo and in vitro populations. As we only had access to samples from a single patient, we did not have a control group to understand the impact of phage application frequency in vivo. However, we were able to simulate

this in our in vitro experiments, where we included two phage treatments: one where phages were added repeatedly (mimicking multiple phage doses during in vivo phage treatment) and one where phage was added once (single dose at the start of the experiment). By comparing these treatments with the in vivo treatment, we were able to gain further insight into how multiple phage doses may affect phage therapy.

## Results

### Resistance and infectivity (co)evolution

#### In vivo

Resistance to phage can evolve via de novo mutations or selection on preexisting resistance and we therefore tested the resistance of ancestral bacterial isolates to ancestral phage and pooled phage isolates isolated in vivo. Of the 24 ancestral isolates isolated in vivo at day 0, 23 were susceptible to both the ancestral phage (14–1 and PNM) while one was resistant to 14–1 but susceptible to PNM. All ancestral bacteria were susceptible to phage isolated from the patient during and after decolonization (*Figure 1a*). We considered whether bacteria and phage coevolved in vivo using time-shift assays to determine changes in resistance and infectivity through time. After 2 days, we saw evidence of resistance evolution, with more than half of bacterial isolates being resistant to ancestral phage. One of the two isolates isolated at day 4 was phage resistant and no *P. aeruginosa* was detected from nasal swabs after this time point. Although both bacterial populations from days 2 and 4 showed observationally higher phage resistance (87.5% day 2, 100% day 4) against day 2 phage, this was nonsignificantly different to resistance shown to ancestral phage and phage from days 4 and 7 (Tukey HSD: p>0.05; *Supplementary file 1*; *Figure 1a*). This nonsignificance is likely due to low statistical power from limited clonal numbers (only eight isolates from day 2) and the consistency of resistance levels in bacterial populations from day 2 (apart from resistance to contemporary phage). Overall, results showed equal levels of phage resistance over time in vivo. The absence of changes in phage infectivity through time did not warrant further investigation of the population dynamics of the two inoculated phages.

#### In vitro

We used the same bacterial isolates and phages to determine how well the in vivo evolutionary dynamics paralleled in vitro experiments in the lab. To assess the impact of phage application schedule, we added phages once at the start of the experiment, repeatedly (as was the case in vivo) or not at all. The dynamics in the phage treatments largely mirrored the in vivo dynamics. Resistance to phage increased through time, reaching comparable levels to that observed in vivo (>50%). While phage resistance increased in both phages treatments (ANOVA comparing models with and without bacterial time: phage added: once $\chi_2^2$=79.08, p<0.001; repeatedly $\chi_2^2$=45.30, p<0.001; *Figure 1b and c*), there was a significantly higher proportion of bacteria resistant to contemporary phage present at the end of the experiment in the treatment where phages were added once (90.81%, ±SE=0.0003) compared to when phages were added repeatedly (66.7%, ±SE=7.64; Tukey HSD comparing phage added once and repeatedly, treatments:estimate=−1.63, z-ratio=−3.75, p=0.0002; *Figure 1b, c*). All isolates from the control condition were susceptible to ancestral phages.

The lower resistance of bacteria when phages were added repeatedly was driven by evolution of increased phage infectivity. When phages were added once, resistance to ancestral phage (14–1: 88.64%, ±SE=2.77; PNM: 90.15%, ±SE=2.60) did not differ relative to phage isolated at the end of experimental evolution (90.0%, ±SE=4.27); Tukey HSD comparing end-point phage resistance to contemporary phage to resistance to 14–1 (p=1.0) and PNM (p=0.938; *Figure 1b*). However, when phages were added repeatedly, resistance to ancestral phage (14–1: 94.41%, ±SE=1.93; PNM: 96.5%, ±SE=1.54) was significantly higher than to phage isolated at the end of experiment (61.5%, ±SE=12.5), indicative of increased phage infectivity (Tukey HSD comparing end-point phage resistance to contemporary phage to resistance to 14–1 (p<0.001) and PNM (p<0.001); *Figure 1c*). We saw no further evidence of phage evolution in either treatments, with phage resistance being independent to the time point from which phage was isolated during experimental evolution (ANOVA comparing models with and without phage time: phage added: once $\chi_2^2$=0.41; p=0.814; repeatedly $\chi_2^2$=0.68; p=0.713; *Figure 1c*). This suggests the increase in phage infectivity evolved before the first

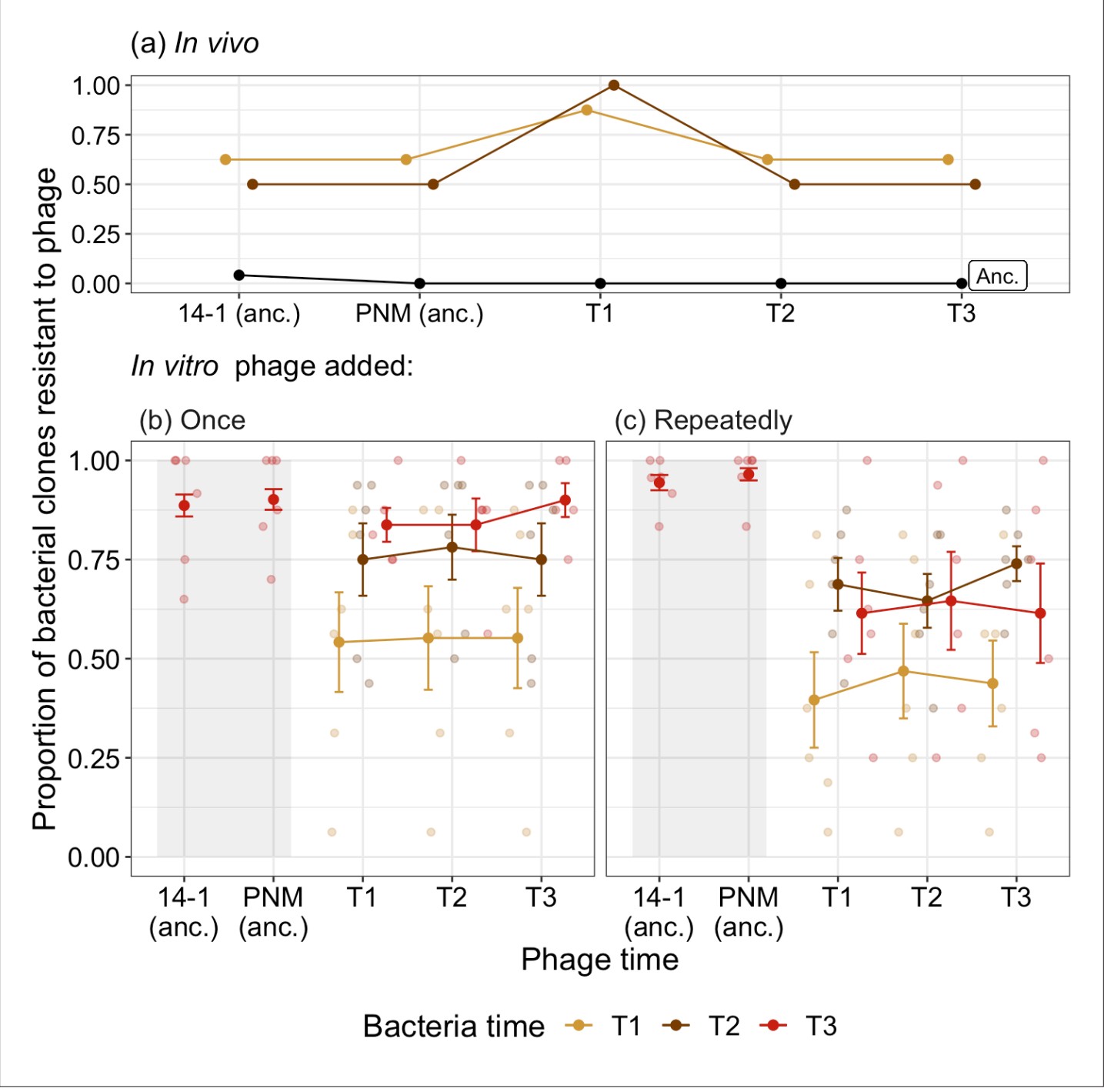

**Figure 1.** Time-shift assays of bacterial evolution and phage during phage therapy. The proportion of phage resistant bacterial isolates from
(**a**) ancestral (pre-phage therapy, "Anc."), and days 2 (T1) and 4 (T2) of in vivo (clinical) phage therapy (n=8) tested against ancestral phage ('anc.': 14–1
and PNM) and phage isolates from days 2 (T1), 4 (T2), and 7 (T3). The same ancestral bacterial isolates are evolved in vitro with phage added either
(**b**) once, at the start of the experiment for comparison to (**c**) in which phage is added repeatedly, emulating clinical treatment. For (**b**) and (**c**), the
proportion of bacterial isolates susceptible to phage at three time points (T1–T3: days 4, 8, and 12 of coevolution) is presented with small points as
independent treatment replicates. Resistance of bacteria isolated at T3 to ancestral phage (anc.: 14–1 and PNM) is indicated within the shaded region,
note that these isolates are distinct from those tested against phage from T1–T3. Filled points are the mean proportion of resistant isolates across
replicates. Bars=±SE.

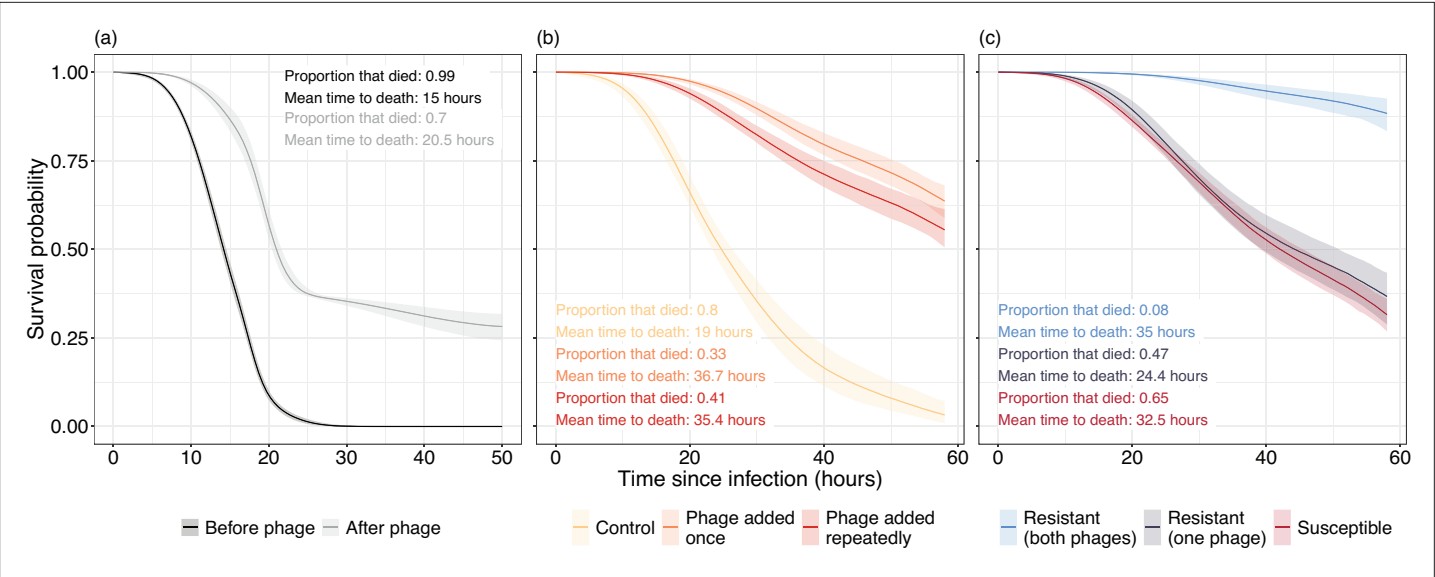

**Figure 2.** Survival curves of *Galleria mellonella* inoculated with *Pseudomonas aeruginosa* isolates isolated (**a**) in vivo and (**b, c**) in vitro. (**a**) Virulence of bacterial isolates isolated from before (black) and after (gray) phage therapy. (**b**) Virulence of bacteria in control populations where no phage is present (yellow) is much greater than populations that were exposed to either a single (orange) or repeated (red) phage applications. (**c**) Virulence is significantly reduced in bacterial isolates resistant to both phages while bacteria resistant to one phage are nonsignificantly different in virulence to phage susceptible isolates. Lines represent the average prediction and shaded bands represent the 95% credible intervals of those predictions. Proportion and mean time to death of *G. mellonella* that died are presented as summary statistics for each treatment.

sampling time point (the second transfer, or 96 hr). The absence of any further changes in phages during experimental evolution did not warrant further investigation of the population dynamics of the two inoculated phages.

## Phage resistance-virulence trade-offs

We next assessed whether phage application altered the virulence of *P. aeruginosa* populations in vivo and in vitro using a *Galleria mellonella* infection model. Here, virulence can be calculated as the inverse of *G. mellonella* survival over time: the more virulent the bacterial isolate, the shorter the time until death. In vivo, phage therapy reduced bacterial virulence by 95.5% on average (median hazard ratio [HR]=0.045, 95% confidence interval [CI]=0.012–0.163). When inoculated with bacteria isolated before phage therapy, 479 (99.7%) of *G. mellonella* died within 46 hr, with death occurring at a mean time of 15 hr (**Figure 2a**). In contrast, after phage therapy, bacteria killed only 70% of *G. mellonella* within 50 hr, with mean time of death occurring at 20.5 hr (**Figure 2a**).

We also saw a reduction in virulence in vitro when populations had evolved with phage compared to without phage. Bacteria in the phage added once and phage added repeatedly treatments were 96.15% (median HR=0.0385, 95% CI=0.011–0.124) and 89.8% (median HR=0.102, 95% CI=0.0294–0.355) less virulent on average compared to the no-phage control populations (**Figure 2b**). In the control populations, 96 (80%) of the *G. mellonella* died within 47 hr, with death occurring at a mean time of 19 hr. In contrast, only 32.5% and 41.2% of *G. mellonella* died in the phage added once and phage added repeatedly treatments, respectively, with a mean time to death of 36.7 and 35.4 hr (**Figure 2b**).

This difference was driven by two impacts of phage application. First, there was a huge reduction in virulence of bacteria that were resistant to both phages. When resistant to both phages, virulence reduced by 95.69% compared to susceptible bacteria (median HR=0.0431, 95% CI=0.0139–0.122) and was 93.33% lower than bacteria resistant to only a single phage (median HR=0.0663, 95% CI=0.0179–0.247; **Figure 2c**). Meanwhile, there was no reduction in virulence for bacteria resistant to just a single phage (median HR=0.653, 95% CI=0.185–2.09) when compared to phage susceptible bacteria. Second, even susceptible bacteria became less virulent after phage application, with a 91.14% (median HR=0.0886, 95% CI=0.0246–0.344) reduction in the phage added once treatment

and an 80.7% reduction when phage was added repeatedly (median HR=0.193, 95% CI=0.04–1.06; *Figure 2c*).

## Growth rate

Evolving resistance against phage can trade-off with growth rate (*Buckling et al., 2006*; *Koskella et al., 2011*; *Wright et al., 2018*), and could be a plausible mechanism for virulence reduction. In the case of the in vivo isolates, the growth rate of pre-phage treatment isolates ($\bar{x}$=0.36, 95% CI=0.342–0.369) was significantly higher than that of post-treatment isolates ($\bar{x}$=0.25, 95% CI=0.195–0.302). Post-treatment isolates which evolved resistance (to one or both phages) had a significantly lower growth rate ($\bar{x}$=0.21, 95% CI=0.162–0.266) than the ancestral isolates ($\bar{x}$ = 0.36, 95% CI=0.342–0.369). However, the 95% CIs of the susceptible isolates' growth rate ($\bar{x}$=0.33, 95% CI=0.256–0.402) overlapped with the ancestral and resistant isolates, indicating nonsignificance, which was likely driven by a lack of statistical power (n=3 susceptible isolates) (*Figure 3a*).

For in vitro isolates, growth rate was significantly affected by treatment ($\chi^2_2$=20.14, p<0.001) with the control group having significantly higher growth rates ($\bar{x}$=0.43, 95% CI=0.373–0.477) compared to treatments in which phage was added once ($\bar{x}$=0.30, 95% CI=0.256–0.346; p=0.002) or repeatedly ($\bar{x}$=0.276, 95% CI=0.235–0.318; p<0.001; *Figure 3b*). No significant difference was observed in growth rates between each phage treatment (Tukey HSD: estimate=0.025, t-ratio=0.95, p=0.628); and changes in growth were not associated with phage resistance (interaction between treatment and resistance: $\chi^2_2$=0.37, p=0.832; fixed effect of resistance: $\chi^2_2$=2.98, p=0.226; *Figure 3b*).

## Biofilm production

Biofilm production is often altered in phage resistant isolates (*Harrison et al., 2015*; *Kim et al., 2015*; *Scanlan and Buckling, 2012*) and this can also affect virulence and growth rate (*Li et al., 2008*; *Wang et al., 2011*). Therefore, we compared the biofilm productivity between isolates evolved in vitro and in vivo. For in vivo isolates, there was no significant difference in biofilm production between isolates isolated pre- ($\bar{x}$=5.53, 95% CI=5.45–5.6) or post-phage therapy ($\bar{x}$=5.39, 95% CI=5.27–5.5), and between phage resistant ($\bar{x}$=5.38, 95% CI=5.27–5.50) and susceptible isolates ($\bar{x}$=5.39, 95% CI=5.1–5.69; *Figure 3c*).

For in vitro isolates, there was no significant difference in biofilm production between treatments (treatment×resistance interaction: $\chi^2_2$=0.785, p=0.675; treatment: $\chi^2_2$=4.52, p=0.104). However, phage resistance was associated with increased biofilm production ($\chi^2_2$=19.12, p<0.001), with phage susceptible isolates ($\bar{x}$=5.35, 95% CI=5.16–5.54) having lower biofilm production than isolates resistant to one ($\bar{x}$=5.85, 95% CI=5.56–6.13; p=0.012) or both phages ($\bar{x}$=5.93, 95% CI=5.73–6.13; p=0.004; *Figure 3d*).

## Genetic basis of evolutionary changes

Given the parallel phenotypic differences we observed under in vivo and in vitro conditions, we next explored whether the underlying genetic changes were the same for in vitro and in vivo evolved populations. Results were not affected whether we included only genetic variants occurring in genes of known function or all genetic variants (*Figure 5—figure supplement 1*). As we were interested in attributing potential functions to the variants identified, we only present the results for genetic variants occurring in genes of known function. From the in vitro pooled sequencing, we found significant changes in 26 different genes with known function (*Figure 4*), 16 of which changed in frequency compared to the ancestral population (selection on standing genetic variation) while 10 were novel mutations (de novo evolution).

In terms of genetic distance from the ancestral population, populations in the treatment where phage was added repeatedly diverged more than when phage was added once (Tukey HSD: t=–4.32, d.f.=15, p=0.0017), which diverged more from the ancestral populations than the control populations (Tukey HSD: t=–8.81, d.f.=15, p<0.0001; *Figure 5a and b*). Consistent with this, more novel SNPs/indels were found in the treatment populations where phages were added repeatedly (mean=19.2) compared to where phages were added once (mean=11.8, Tukey HSD: t=–4.80, d.f.=15, p=0.0006), which in turn had more novel SNPs/indels than control populations (mean=2.5, Tukey HSD: t=–6.105, d.f.=15, p=0.0001; *Figure 5b*). This suggests novel SNPs/indels were potentially more important determinants of phage resistance than selection acting on standing variation. Within-population

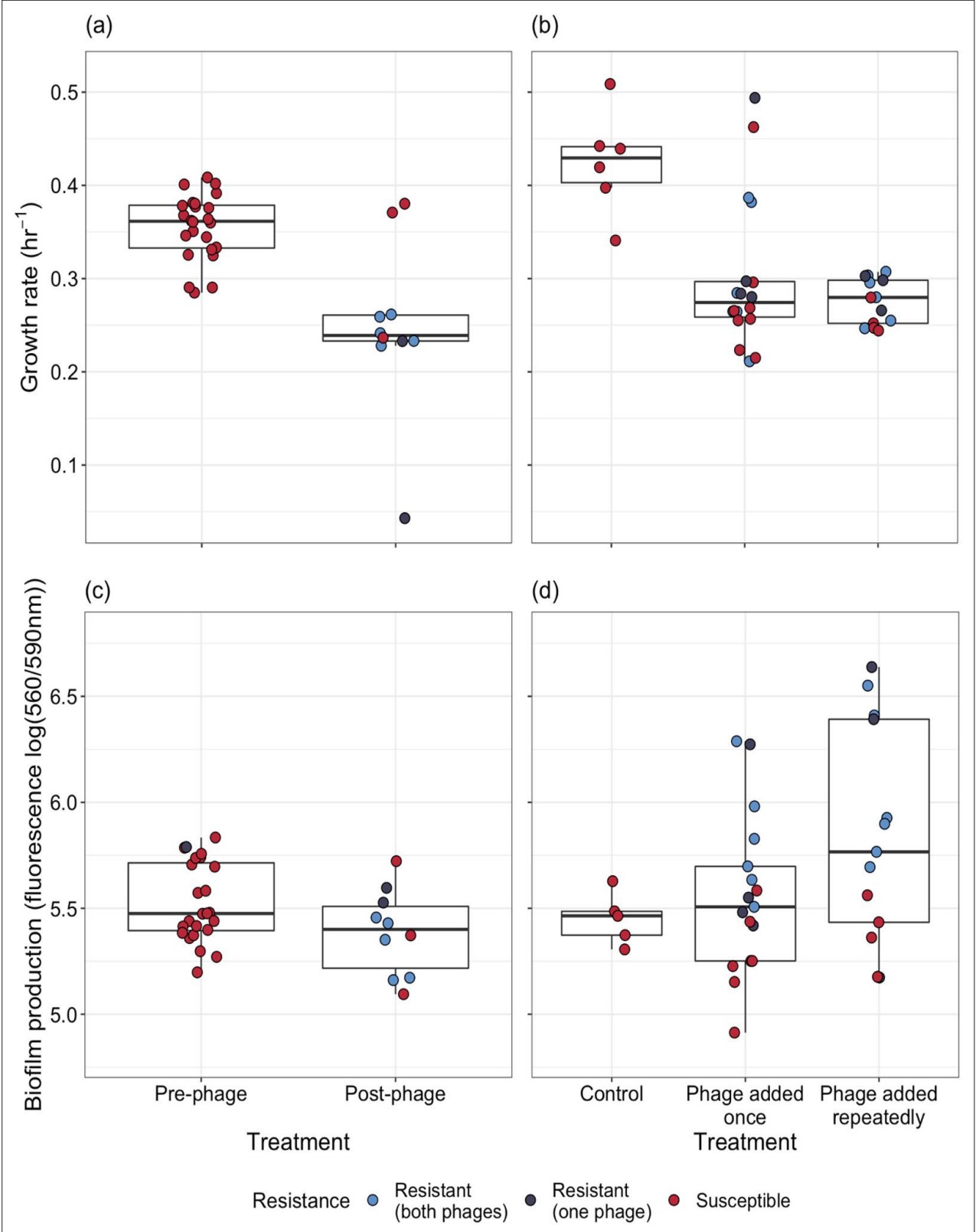

**Figure 3.** Phenotypic changes underpinning the resistance—virulence trade-off of bacterial populations. Populations were evolved (**a, c**) in vivo and (**b, d**) in vitro. (**a, b**) Changes in bacterial growth were estimated for populations isolated (**a**) before and after phage treatment, and (**b**) between control and the phage treatment groups. Single points represent individual bacterial isolates of differing phage resistance levels: resistant to both phages (light blue), resistant to 14–1 or PNM (dark blue), or phage susceptible (red). Tops and bottoms of the bars represent the 75th and 25th percentiles of the data,

*Figure 3 continued*

the middle lines are the medians, and the whiskers extend from their respective hinge to the smallest or largest value no further than 1.5* interquartile range.

The online version of this article includes the following figure supplement(s) for figure 3:

**Figure supplement 1.** Growth curves of bacterial clones as measured by change in log-transformed ocular density ($OD_{600}$) over time (hr).

diversity was highest in phage added repeatedly treatment (Tukey HSD phage added repeatedly vs. phage added once: t=–5.96, d.f.=15, p=0.0001; Tukey HSD phage added repeatedly vs. control: t=–4.33, d.f.=15, p=0.0016), but there was no difference between control populations and phage added once treatment (Tukey HSD: t=–1.14, d.f.=15, p=0.2625; *Figure 5c*).

In vitro treatment significantly altered genetic composition (i.e., the centroids of phage added once, phage added repeatedly, and control populations were different, *Figure 4*, PERMANOVA, $F_{3,15}$=16.05, $R^2$=0.76, p=0.0001). There was no difference in beta-diversity (calculated from distance-to-centroids between groups) between treatments (homogeneity of multi-variation dispersion ANOVA: $F_{3,15}$=1.65, p=0.22). Genetic changes associated with both in vitro phage treatments, but not control, were flagellar hook-associated protein (*flgK*) and rod-shape determining proteins (*mreC* and *mreD*) (*Figure 4*). Additionally, genetic changes in DNA mismatch repair (*mutS*), elongation factor G (*fusA*), signal recognition particle (*ffs*), type three secretion system pilotin (*yscW*), and pilus assembly protein (*pilB*) were mainly found in vitro in the phage added repeatedly treatment (*Figure 4*).

We examined whether these same mutations arose in the clonal in vivo isolates. Out of the 50 genetic changes identified in vitro, the in vivo isolates had a mean of 11.8 changes (maximum=23, minimum=1). Changes in frequencies or mutations were found at 16/26 of the same genes identified in vitro in at least one isolate (*Figure 4*). Crucially, some of the de novo mutations that were observed at high frequency in vitro also had high frequency in vivo (flgL=30%, mutS=45%, mreC=50%, mreD=30%, and pilB=40%). In total, 31 out of 50 genetic changes (62%) in vitro were found in vivo. Of these, 77% were de novo mutations in vitro, indicating that de novo mutations were also important for phage resistance in vivo.

We next assessed whether mutations arising in individual isolates in vivo could be linked to phenotypic changes (resistance, virulence, biofilm productivity, and growth rate). If strong genotypic—phenotype links can be made, we would expect isolates to cluster according to their trait value. However, we found no obvious genotype to phenotype links in vivo and due to the small number of isolates, we were unable to conduct robust statistical analyses (*Figure 5*; *Figure 5—figure supplement 2*).

## Discussion

Parallel evolutionary dynamics were found in vivo and in vitro, showing that laboratory studies can be predictive of certain phenotypic outcomes of clinical phage therapy (phage-mediated decolonization). Levels of phage resistance were comparable in vivo and in vitro with limited phage infectivity evolution occurring only in vitro. While bacteria went extinct in vivo, they persisted with phage in vitro, suggesting that in vitro experiments may be more important in predicting evolutionary than clinical outcomes. Additionally, phage resistance was associated with trade-offs with virulence and bacterial growth and similar mutations or genetic variants could explain these parallel evolutionary changes in both ecological contexts.

Parallel evolution is promising for future clinical work in which the trajectory of bacteria and phage (co)evolution may be predicted for individual patients. Given the difference in environments between that of a laboratory and human nasal cavity, it was surprising that evolution was so similar. The nasal cavity would have been more nutrient limited and spatially heterogenous and such conditions have been shown to change bacteria-phage interactions compared to evolution in high-nutrient laboratory growth medium (e.g., Luria broth [LB]). For instance, bacteria-phage coevolution is altered in soil compared to growth medium because of increased costs of resistance in the former (*Gómez and Buckling, 2011*); and in plant leaves, bacteria and phage have failed to evolve while otherwise coevolving in the laboratory due to lower bacteria—phage contact rates (*Hernandez and Koskella, 2019*). Other media types can be used in vitro to replicate in vivo conditions more close (e.g., sputum

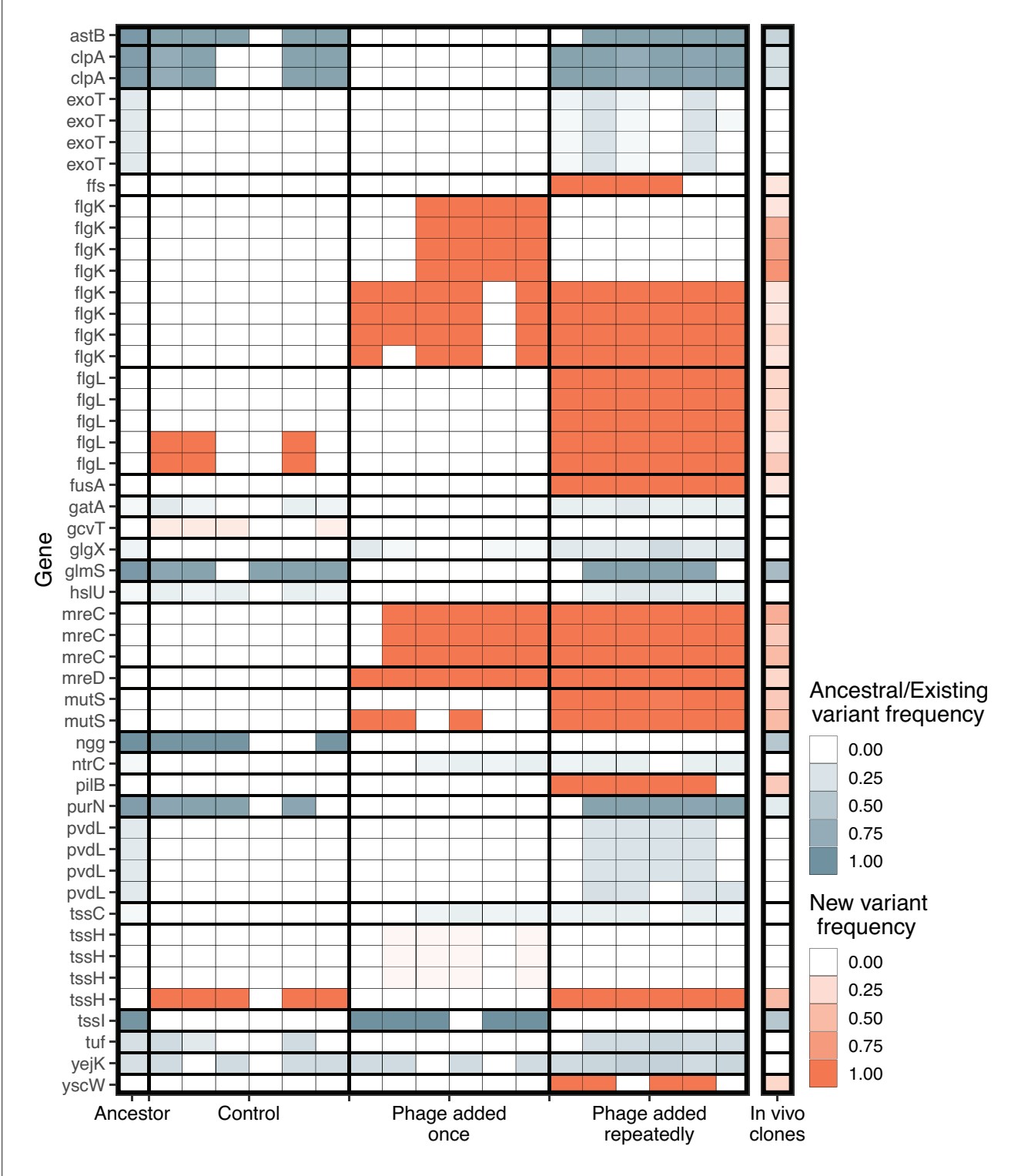

**Figure 4.** Significant genetic changes identified in vitro and in vivo. The frequency of genes that differed significantly between treatments in vitro is shown. Genetic changes that were already present in the ancestral population (first column) are shown in blue, while de novo mutations are in orange. Only genes whose function was identified are included. Multiple rows of the same gene indicate different genomic variants of the same gene. Colour intensity shows how frequent the genetic change is in the population. For in vitro treatments, each column is an independent replicate representing the results of the pooled sequencing. For the in vivo isolates (final column), all isolates were pooled to visualize the frequency of each genetic change in vivo.

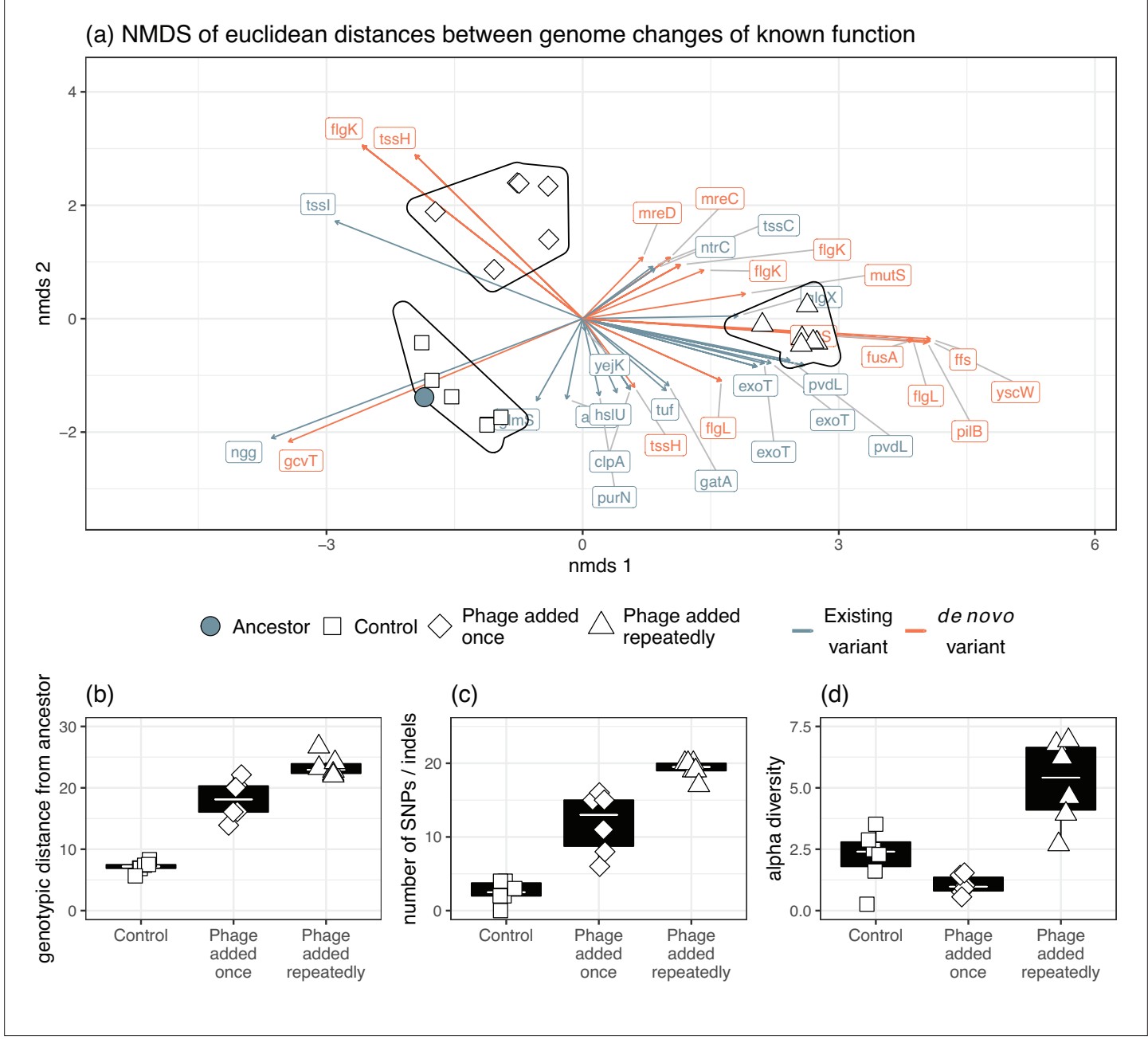

**Figure 5.** Genomic changes for populations evolved in vitro. (**a**) The divergence of each treatment group as indicated by Euclidean distances of each gene from the ancestral isolates. Genetic changes were characterized as a change in frequency of an existing gene in the ancestor or selection on a mutation that was gained during experimental evolution. The presence of phage significantly affected (**b**) genotypic distance from the ancestor, (**c**) the number of new SNPs, and (**d**) alpha diversity.

The online version of this article includes the following figure supplement(s) for figure 5:

**Figure supplement 1.** Genomic changes for populations evolved in vitro, including genetic variants occurring in genes of known and unknown function.

**Figure supplement 2.** Associations between genetic similarity and phenotypic traits for in vivo clones.

media) than LB used here which may be more appropriate for other infection types such as those infecting lungs of cystic fibrosis patients (*Palmer et al., 2007*; *Palmer et al., 2005*).

Rapid phage resistance has been highlighted as a concern for phage therapy development (*Torres-Barceló, 2018*) and was found in this case study. Without ongoing reciprocal phage evolution, phage resistance is likely to lead to regrowth of bacteria during infections; this has been found in a few case

studies but has been combatted by selecting a different phage (single or cocktail) (*Aslam et al., 2020*; *Schooley et al., 2017*; *Wu et al., 2021*; *Zhvania et al., 2017*). That many de novo mutations—as well as selection on existing variation—occurred during bacteria-phage coevolution in vitro and many were also found in vivo (77% of variants observed in vivo also evolved de novo during in vitro coevolution) further highlights the fact that stressful in vivo conditions do not prevent bacteria evolving resistance to phage. However, phage resistance was found to have growth rate costs which may further explain why some isolates remained sensitive to phage, and why decolonization was ultimately successful. The natural nasal microbiota, and host immunity, may have further contributed to this success by outcompeting slower growing phage resistant isolates (*Wang et al., 2019*). Bacteria did not go extinct in our in vitro experiments which may be reflective of differences in environmental conditions such as nutrient availability and interspecies competition.

Resistance-virulence trade-offs have been found in previous in vitro animal models and clinical studies and are expected to improve treatment success in spite of phage resistance (*Oechslin, 2018*). Phage resistance however is not always costly and likely depends on the genetic background of the bacteria and phage, as well as the environmental context (*Meaden et al., 2015*; *van Houte et al., 2016*). The absence of clear links between resistance phenotypes and growth rate within our in vitro treatments, despite clear between-treatment effects, is presumably because the cultures were initiated with genetically diverse clinical isolates, rather than a single ancestral clone. Ultimately, phage resistance was beneficial in controlling the pathogen in this case study and such benefits could have led to treatment success.

Population genomics of the isolates in vivo and in vitro revealed parallel genomic changes and the effects of phage selection at a molecular level. In line with previous studies, phage selection accelerated molecular evolution which resulted in populations being more diverged from their ancestral pool than control isolates (in vitro) (*Betts et al., 2018*; *Morgan et al., 2010*; *Pal et al., 2007*). This suggests that phage placed strong selection on bacteria and this further resulted in mutator strains (*mutS* genotype), especially where phage was added repeatedly in vitro and in vivo, that likely accelerated the rate of evolution further (*Morgan et al., 2010*). Between both in vivo and in vitro results, mutations selected by phage gave insight into potential mechanisms of resistance. In particular, mutations to *mreD* and *mreC* were encoded for cell shape (*Cabeen and Jacobs-Wagner, 2005*) while missense mutations at *flgK* and *flgL* probably resulted in flagella loss (*Macnab, 2003*). The result of these mutations was likely reduced cell surface area which reduces the probability of phage absorption (*Dennehy and Abedon, 2021*). *pilB* mutations were also selected which is likely a surface mutation to prevent PNM (a pilus binding phage) from attaching (*Ceyssens et al., 2011*).

Strong genotype-phenotype links were difficult to distinguish as a high number of the same mutations or existing variants were selected upon both in vivo and in vitro, and no unique candidate genes were obvious to explain phenotypic differences (i.e., growth rate and biofilm production). Flagella and *mreC* mutations are associated with increased biofilm production (*Whiteley et al., 2001*) and as they were found in the in vitro isolates, this may suggest a generalistic phage defence strategy (*Harper et al., 2014*). However, as these same mutations were found in vivo, without increases in biofilm production, strong genotype—phenotype links are unclear. Surface mutation is a common form of phage resistance (*van Houte et al., 2016*) but here it appears coupled with mechanisms that prevent absorption more generally; combined, these strategies could have reduced bacterial growth and/or virulence by impairing swarming motility (*Haiko and Westerlund-Wikström, 2013*). Motility (*Alseth et al., 2019*) and resource-use assays (using Biolog plates) may have provided some stronger genotype-phenotype correlations, but this was beyond the scope of this work. In contrast, other studies using the same phage but *P. aeruginosa* PA01 instead reported mutations predominantly at pilus and LPS receptors, showing that greater selection is placed on specific phage resistance strategies at phage binding sites (*Wright et al., 2019*; *Wright et al., 2018*). Our results highlight the importance of genetic background, and hence the use of clinical isolates over laboratory clones, in determining evolutionary trajectories (*Betts et al., 2014*; *Friman et al., 2016*; *León and Bastías, 2015*; *Rossitto et al., 2018*; *van Houte et al., 2016*).

How phages were applied in vitro had an impact on phage resistance. Although we observed a greater number of mutations where phages were added repeatedly (mimicking multiple phage doses), the proportion of resistant bacteria (resistance to contemporary phage) at the end of the experiment was significantly lower (61.5%) compared to where phages were just added at the start

of the experiment (90%). That resistance of evolved bacteria to the ancestral phages (~90%) did not differ between treatments suggests that differences in resistance to evolved phages were caused by phage infectivity evolution in the repeated phage treatment. This infectivity evolution occurred before the first time point, with no subsequent changes, and was presumably the result of greater mutation supply rate in this repeated phage application treatment. It is interesting that although this 'repeated' treatment mimics clinical methods more closely, these same patterns were not observed in vivo. However, as bacteria went extinct in vivo, it is likely phage generations were fewer which may have restricted phage evolution in the lower resource environment (*Lopez-Pascua and Buckling, 2008*). Multiple (*Manohar et al., 2018*) and higher (*Barrow et al., 1998*; *Biswas et al., 2002*; *Debarbieux et al., 2010*; *Merril et al., 1996*; *Sunagar et al., 2010*; *Vinodkumar et al., 2008*; *Wang et al., 2006*; *Wills et al., 2005*) phage doses have been shown previously to decrease morbidity and mortality in animal models. Our results build upon this work to show that maintaining high phage titers may have a positive effect in reducing resistant populations despite increasing the strength of selection for resistance.

Although our results suggest that the results of laboratory studies may parallel phage resistance—virulence trade-offs during in vivo phage therapy, there a number of important caveats. First, our in vivo results are based on that of a single patient which limits the generalizability of our results. Nevertheless, our replicated in vitro experiments agreed with our in vivo findings, which provide strength behind our conclusions. Second, our in vitro—in vivo results were comparable with regards to the direction of change, but absolute effect sizes were greater in vitro. For instance, we observed a higher proportion of phage resistant isolates in vitro that were also far less virulent than isolates found in vivo. Therefore, it may be expected that phenotypic changes will be exaggerated in vitro. These parallels were further dependent on the different underlying mechanisms (biofilm formation or growth rate) mediating the resistance-virulence trade-off having the same knock-on phenotypic outcomes. Had different environments selected for different phage resistance strategies, this may have resulted divergent phenotypic outcomes for virulence (*Alseth et al., 2019*). Additionally, as this patient had a nasal colonization, the role of the adaptive immune system would have been heavily reduced (*Ooi et al., 2008*). The immune system has the potential to improve phage therapy success by synergistically killing bacteria with phage which may diminish the potential for phage resistance by reducing bacterial growth (*Leung and Weitz, 2017*; *Roach et al., 2017*). Alternatively, immune cells may kill phage and therefore prevent phage acting on bacterial populations (*Van Belleghem et al., 2018*). Additionally, the nasal cavity has a diverse microbiome that may have contributed to *P. aeruginosa* decolonization (*Hu et al., 2018*; *Kumpitsch et al., 2019*). The role of the microbiome will differ in infection type, with gut and respiratory infections potentially having a greater microbiome effect (*Hu et al., 2018*; *Kumpitsch et al., 2019*; *Lloyd-Price et al., 2016*) while septic infections may have a greater immune system effect (*Van Belleghem et al., 2018*). Therefore, it is important to compare laboratory vs in vivo phage therapy in a number of infection types. The genomic background of the bacteria may further affect how sensitive its resistance strategies are to the environment it is in as more costly resistance strategies are more likely to be selected in vitro than in vivo (*van Houte et al., 2016*). This encompasses not only genetic variation within species but also differences between species that may differ in infection sites, immune responses, virulence and sensitivity to phage (*Abedon, 2019*; *Doffkay, 2015*; *van Houte et al., 2016*). Further work using isolates from case studies and clinical trials will give an insight into the predictability of bacteria and phage evolution from laboratory to in vivo studies.

## Materials and methods
### Patient information, bacteria, and phage strains and phage-bacteria evolution in vivo

Bacterial isolates were sampled from a patient as part of a clinical trial in Brussels, Belgium. The patient (85 years old, male) had been admitted to an ICU in 2015 for treatment of burns (10% body area). Independent of his burns, a nasal colonization of *P. aeruginosa* was observed. Here, the patient was treated every day with a three-phage cocktail (BFC-1) for 5 days, which consisted of two phages (14–1 and PNM) that could lytically infect *P. aeruginosa*. The phage cocktail included a third phage that specifically infects *S. aureus* bacterium, but this bacterium was not detected in this patient. The phage cocktail has previously been characterized to ensure no prophage elements are present or

that the phage carry toxin-encoding genes that may increase bacterial resistance and/or virulence (*Merabishvili et al., 2009*). Furthermore, a purification process including the removal of endotoxins in phage production was used to ensure such waste products cannot harm patients (*Merabishvili et al., 2009*). As 14–1 infects via a lipopolysaccharide (LPS) (*Betts et al., 2014*; *Kropinski et al., 1977*) receptor and PNM uses type IV pili (*Ceyssens et al., 2011*), resistance to one phage is unlikely to result in cross-resistance and neither phage should compete for receptor sites (*Wright et al., 2018*). Swabs of both nasal cavities were taken on day 0 (commencement of phage therapy), 2, 4, 7, and 13 and frozen at –80°C in lysogeny broth containing 20% glycerol. Individual *P. aeruginosa* isolates were obtained by plating swabs onto cetrimide selection plates. Colonies were picked and inoculated into LB and grown overnight, shaking at 180 rpm, at 37°C to achieve high cell densities. Additionally, swabs were soaked in LB broth to amplify phage overnight in the same conditions. Phage was isolated from overnight cultures via chloroform extraction: 100 µl of chloroform was added 900 µl of culture, vortexed, and then centrifuged (14,000 rpm, 1 min); the supernatant was isolated and stored at 4°C. Culture samples were frozen in 80% glycerol (final concentration 40%) at –80°C. Spot assays on lawns of *P. aeruginosa* P573 (a phage susceptible control strain) confirmed the presence of phage at all time points. Twelve individual plaques were picked at each time point and amplified with an overnight culture of P573 to obtain clonal isolates. Individual isolates of *P. aeruginosa* were isolated from days 0 (n=24), 2 (n=8), and 4 (n=2) for use in this study and swabs were negative on days 7 and 13.

## Experimental in vitro phage-bacteria evolution

To examine whether findings in vitro are consistent with in vivo bacteria-phage (co)evolution, phage therapy was replicated in the laboratory. We used two treatments: (1) phage added 'once', at the start of the experiment, and (2) phage added at every second transfer ('repeated' treatment). The repeated phage treatment aimed to emulate the conditions of the clinical trial (while preventing the buildup of too many ancestral phages in the 'closed' in vitro system) and the 'once' condition was used to assess whether repeated doses change the observed dynamics. The phages were identical to the two lytic phages (14–1 and PNM) used in the clinical trial. A control group in which bacteria evolved in the absence of phage was set up and there were six biological replicates in the control and each phage treatment.

Prior to experimental setup, in vivo ancestral isolates (n=24, same isolates as day 0) were individually amplified in LB media and cultured at room temperature for 2 days. Cell densities were estimated via optical density (wavelength 600 nm; $OD_{600}$) measures. Cultures were centrifuged (14,000 rpm, 3 min), the supernatant removed, and resuspended in M9 buffer to equalize cell densities. These were added to a master-mix in which isolates were equally represented in equal ratio (total $10^8$ CFUs/ml; ~$4\times10^6$ CFUs/ml of each isolate). 100 µl of master-mix ($10^7$ CFUs; ~$4\times10^5$ CFUs per isolate) was inoculated into 5 ml LB media. 100 µl of phage was added to treatment microcosms at a total multiplicity of infection (MOI) of 0.5. Microcosms were cultured, shaken (180 rpm), at 37°C. 1% transfer (50 µl culture into 5 ml LB) took place every 2 days for a total of 12 days. At each transfer, culture samples were frozen in glycerol, and in the phage treatments, phages were extracted as described previously. Cultures were diluted and plated from frozen stocks onto LB agar and incubated for 2 days at 37°C. From each biological treatment replicate and time point (days 4, 8, and 12), 16 bacterial colonies were picked (96 total colonies) for time-shift assays to measure coevolutionary changes. An additional 24 isolates were isolated at day 12 from each treatment and control group replicate for bacterial virulence, growth rate, and biofilm assays. Picked isolates were individually amplified in LB media overnight before being frozen as described previously.

## Measuring resistance-infectivity (co)evolution

To assess whether bacteria and phage had coevolved in vivo and in vitro, resistance assays were conducted where bacteria were exposed to phage populations from the same/contemporary, past, and future time points (time-shift assays) (*Buckling and Rainey, 2002*). If bacteria and phage coevolved, both bacteria and phage would show significant changes in resistance and infectivity, respectively, through time. Increases in resistance and infectivity would be indicative of arms race dynamics, while nondirectional changes would be indicative of fluctuating selection dynamics (*Lopez Pascua et al., 2014*). Infectivity/resistance was determined by spot assays in which 100 µl of overnight bacterial cultures was added to 5–8 µl soft LB agar overlay and 5 µl of each ancestral phage, or evolved phage

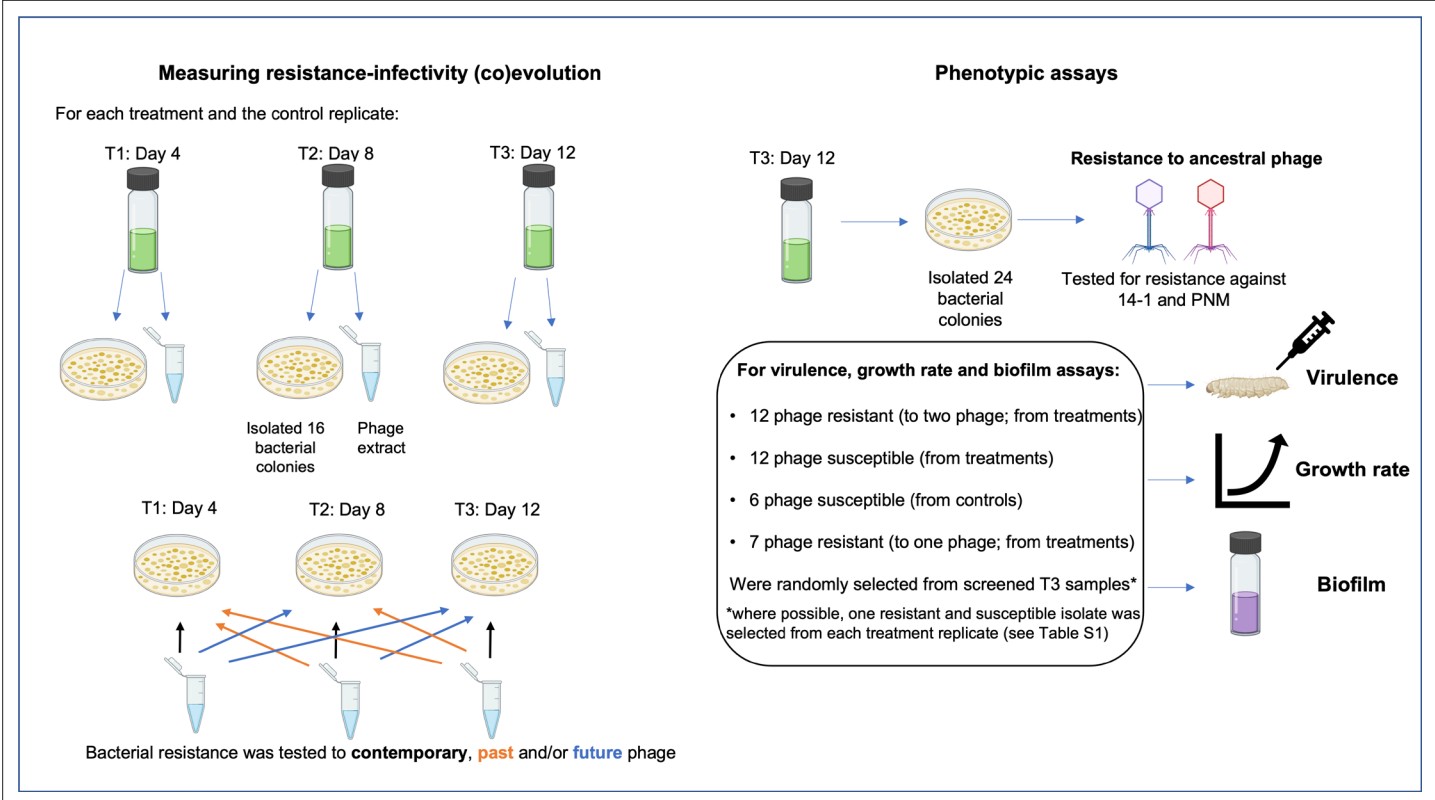

**Figure 6.** Selection process of colonies used for time-series assays measuring resistance-infectivity (co)evolution and phenotypic assays (end-point resistance, virulence, growth rate, and biofilm) for isolates evolved in vitro. For time-series assays, 16 bacterial colonies were randomly selected from each treatment replicate (phage added once or repeatedly, n=6 each) and time point (T1–T3), alongside a phage extract that contains a pool of phage genotypes and species (14–1 and PNM). Each colony was then tested for resistance to the pooled isolated across each time point. For phenotypic assays, 24 colonies from T3 replicates of each treatment were screened for resistance to each phage. From these 24 colonies, bacterial colonies of each resistance type were randomly selected for further phenotypic assays. Created with icons from BioRender.com.

mixture, was spotted onto the overlay. For in vivo isolates, ancestral isolates (isolated pre-phage therapy) and bacterial isolates from days 2 to 4 were tested against the phage clonal isolates from days 2, 4, and 7. A bacterial isolate was scored as susceptible (1=susceptible, 0=resistant) if at least 1 of the 12 clonal phage isolates could infect. For in vivo analyses, each isolate was treated as an individual biological replicate. In vitro isolates from days 4, 8, and 12 were tested (16 colonies per time point) against phage populations isolated from each time point (*Figure 6*). Each treatment replicates from each time point were defined as biological replicates, while clonal isolates are biological replicates nested within each treatment replicate. In addition, we assessed the infectivity of both ancestral phage against the ancestral bacteria (n=24), the in vivo isolates (n=8), and the in vitro populations from day 12 (n=24 per treatment replicate; *Figure 6*).

## Virulence assays

Changes in bacterial virulence were measured using a *G. mellonella* infection model (*Tsai et al., 2016*). Although lacking an adaptive immune system, *G. mellonella* innate immune systems are comparable with that of vertebrates, making them suitable, and well-used models of virulence in acute infections (*Tsai et al., 2016*). Note that we define virulence as increased host mortality resulting from parasite infection (*May and Anderson, 1990*). Virulence of all ancestral (n=24) and in vivo isolates (n=10) were measured; however, two in vivo isolates (from day 2 of phage therapy) were excluded from analyses as time of death was not accurately determined. As few isolates remained phage susceptible (n=3), these were pooled with phage resistant isolates to determine a change in virulence from pre- (ancestral, n=24) to post-phage treatment (n=8). For in vitro isolates, where possible, one resistant and one susceptible (to ancestral phage) isolate were selected from each treatment replicate to give a total of

12 resistant and 12 susceptible isolates, with 14 isolates from the 'phage added once' and 10 isolates from the 'phage added repeatedly' (*Figure 6*). As some treatment replicates contained no susceptible isolates, multiple susceptible isolates were chosen from the same replicate (*Supplementary file 2*). Seven in vitro isolates evolved resistance to one phage only (single resistance), therefore their virulence was measured to assess whether changes in virulence were mediated by resistance to one or both phages. The virulence of in vitro isolates was contrasted to six isolates from the control group isolated from six independent replicates.

For each bacterial isolate (total of 56 biological replicates), 20 *G. mellonella* larvae (20 technical replicates) were infected with 10 µl (~100 CFUs) of overnight culture (cells resuspended in M9 buffer) using a sterile syringe. Twenty *G. mellonella* larvae were injected with 10 µl of M9 buffer as a technical control. Survival of each larva was assessed every 2 hr for up to 58 hr.

## Growth rate assays

We measured bacterial growth rates as a potential mechanism underlying virulence variation and cost of resistance for ancestral and all evolved isolates (in vitro and in vivo) tested in the virulence assay. Growth curves were measured in a 96-well plate with 10 µl of culture ($1.0 \times 10^5$ CFUs) from each isolate inoculated into individual wells containing 140 µl LB broth. Twenty-six wells remained as blank controls. The plate was incubated in a spectrophotometer (Biotek Instruments Ltd) at 37°C in shaken conditions and $OD_{600}$ was measured every 20 min for 24 hr to estimate changes in cellular density. Growth curves were independently replicated three times (technical replicates).

## Biofilm assay

Biofilm production of isolates isolated for virulence and growth rate assays was measured using a resazurin assay, based upon a previously described protocol (*Peeters et al., 2008*). Overnight mono-cultures were diluted to 0.1 $OD_{600}$ (~$10^9$ CFU/ml) into M9 buffer. 100 µl of each culture was inoculated into a sterile 96-well plate and incubated at 37°C for 4 hr for cellular adhesion. Then, the supernatant was removed, and wells were washed gently with 100 µl of M9 buffer by pipetting up and down three times. 100 µl of fresh LB media was added to each well including 25 controls (no bacteria) and plates were incubated for 16 hr. Liquid was removed and wells were washed with 100 µl M9 buffer. 100 µl of fresh M9 was added followed by 10 µl resazurin solution (final resazurin concentration: 100 µM). Fluorescence ($\lambda$ex: 560 nm and $\lambda$em: 590 nm) was measured after 1-hr incubation. Biofilm assays were independently replicated three times (technical replicates).

## Sequencing

We sequenced the genomes of the bacterial isolates used for the in vitro experiment to establish to what extent evolution was driven by de novo mutation or selection on existing genetic variation. This is especially pertinent as the experiment did not start from a clonal population, rather from 24 isolates (which included an isolate that was resistant to one phage). At the end of the experiment, we picked 24 isolates, extracted DNA from isolates individually, and performed whole-genome sequencing (WGS) on pools of the bacterial isolates that were isolated from each replicate (pool-seq). Total DNA extraction (19 pooled-isolate samples) was performed using the 'Qiagen QIAmp DNA Mini Kit' following the manufacturer's instructions. An Illumina HiSeq 2000 sequencer was used to generate 250-bp paired reads from a 500-bp insert library.

Reads were trimmed for the presence of Illumina adapter sequences and for quality scores below 10 using *TrimGalore* (v0.6.4_dev, powered by *Cutadapt*). *TrimGalore* also removed any paired-end reads when one of the paired-end reads was shorter than 20 bp. Trimmed reads were mapped to the *P. aeruginosa* UCBPP-14 reference genome (Locus: NC_008463) with *bwa-mem* (v0.7.17-r1188), and duplicates were marked using *samblaster* (v0.1.24). Subsequent bam files were split into mapped and unmapped reads, sorted, and indexed using *samtools* (v0.1.19-44428cd). Across all files, ~93% of the reads mapped to the reference genome (minimum 92.5%), giving on average ~12,000,000 reads per replicate (minimum 8,807,569). Variants were identified using *freebayes* (v1.3.2-dirty) with ploidy set to 24 (*-p 24*) and we assumed pooled samples with a known number of genome copies (*--pooled-discrete*). Due to the high ploidy of the pooled samples, we reduced the number of alleles that are considered at the cost of sensitivity to lower frequency alleles at multiallelic loci by setting *--use-best-n-alleles 4*.

Next, we compiled the vcf files (standard output of SNP callers) into *R* using the package *vcfR* and kept only the variants that were sequenced at a depth within two standard deviations of the mean. Variants were further filtered to retain only those with quality scores >30. We used the two replicate sequencing runs of the same samples to conservatively identify variants; variants were only kept if they occurred in both sequencing runs and if there was less than 0.3 difference in frequency between runs. We then only kept variants that had different frequency than the ancestor (>1/24 difference in frequency). To identify significant differences in genetic variants (SNPs and indels) between treatments, we first filtered unique that were present in at least three replicates of any treatment. This identified 960 potential variants. We then performed a Wilcoxon test on each variant, with frequency as the response variable and treatment (control, phage added once, and phage added repeatedly) as the predictor. Significant genetic variants were identified if the adjusted p value (*fdr* method) was <0.05. Of these significant changes, 160 were existing genetic variants and 120 were novel mutations. We then checked where each variant was in the genome and only retained variants that occurred in a gene. After these steps, 50 genetic variants were identified in genes of known function, and 236 in genes of unknown function. We performed all downstream statistical analyses on (a) only genetic variants in genes of known function and (b) all genetic variants.

For the in vivo isolates, DNA extraction and WGS were conducted on each isolate individually by MicrobesNG using in-house protocols (*MicrobesNG Sequencing Service Methods, 2021*). WGS was carried out on each isolate individually by MicrobesNG using Illumina sequencers (HiSeq/NovaSeq) to create 250-bp paired-end reads. In-house processing by MicobesNG trimmed the adapters and removed reads with a quality <15. We then processed the reads as for the pooled sequencing, except ploidy was set to 1 when using *freebayes*. On average, 81% of reads mapped back to the reference genome, but there was more variation as compared to the in vitro experiment, with a minimum of 44% and a maximum of 94%. As we only had isolates from a single patient, there was limited statistical analysis we could do, so we primarily concentrated on whether the significant changes found in vitro were also present in the in vivo isolates.

## Statistical analyses

All data were analyzed using R (v.4.0.3) in RStudio (*R Development Core Team, 2013*) and all plots were made using the package '*ggplot2*' (*Wickham, 2016*). Model simplification was conducted using likelihood ratio tests and Tukey's post hoc multiple comparison tests were used to identify the most parsimonious model using the R package '*emmeans*' (*Lenth, 2018*). First, we established whether bacteria and phage had coevolved in vivo and in each in vitro treatment using general linear mixed-effects models. Models were fitted using the package '*lme4*' (*Bates et al., 2015*). In vivo isolates from day 4 were excluded as their small sample size (n=2) prevented model convergence. For in vivo isolates: infection (1=susceptible, 0=resistant) was analyzed against phage time point (fixed effect) with a binomial error structure. A random effect of 'isolate' was included to account for the same bacterial isolate being used in infectivity assays across each phage time point. For in vitro analyses, the proportion of isolates infected was analyzed against fixed effects of phage and bacterial time points with separate models for each phage treatment. To see if there were any differences in phage resistance to ancestral phage or in vitro phage isolates at the end of the experiment, the proportion of isolates infected was analyzed against fixed effects of phage treatment and phage type (ancestral or end-point phage). All in vitro models included a random effect of 'replicate' (which treatment replicate isolates had been isolated from) with a binomial error structure.

Virulence of bacterial isolates was analyzed based on survival of *Galleria* infected with different bacterial isolates. Survival curves were fitted using Bayesian regression models using the package '*rstanarm*' (*Goodrich, 2020*) and parameters were estimated using '*tidybayes*' (*Kay and Mastny, 2020*). For all models, we fit a proportional hazards model with an M-splines baseline hazard. Models were ran for 3000 iterations and three chains were used with uninformative priors. Model convergence was assessed using Rhat values (all values were 1) and manually checking chain mixing. For all models, log hazards were estimated for each treatment value and HRs were calculated as the exponential of the difference between two log hazards. A HR below 1 indicates a decrease in virulence compared to the baseline treatment, with a value above 1 indicating an increase in virulence. Median HRs with 95% credible intervals that do not cross 1 indicate a significant difference in virulence between the two

factors. Proportion of *Galleria* that died and mean time to death of *Galleria* that died were calculated as summary statistics for each treatment.

For virulence analyses of in vivo isolates, treatment (pre- or post-phage therapy) was the only predictor of survival, with a random effect of 'isolate' added to account for multiple *Galleria* larvae being infected by the same isolate. For the in vitro isolates, survival was analyzed against interacting fixed effects of resistance (resistant, single resistant, and susceptible) and treatment (no phage added, phage added once, and phage added repeatedly). Here, an additional random effect of 'replicate' was included to account for clonal nonindependence.

Biofilm productivity of in vitro isolates was estimated using linear mixed-effects models. Biofilm production (fluorescence 560/590 nm) was log transformed to account for the data being abundance-based. Biofilm production was averaged over each technical replicate for each biological replicate (individual bacterial isolates). Biofilm production was used as the response variable and explained with interacting fixed effects of treatment and phage resistance, while treatment replicate was included as a random effect. Due to the low number of isolates, biofilm productivity of in vivo isolates from before and after phage treatment as well as phage susceptible and resistant isolates were estimated using a nonparametric bootstrap using the R package '*boot*' (*Canty and Ripley, 2021*). Non-overlapping 95% CIs were interpreted as being significantly different.

Growth rates were estimated using a rolling regression taking the steepest slope of the linear regression between *ln* $OD_{600}$ and time in hours in a shifting window of every seven time points (every 2.3 hr). Growth rate was averaged over each technical replicate for each biological replicate (individual bacterial isolates). For in vitro isolates, growth rate was analyzed against treatment and resistance in a linear mixed-effects model with a random effect of treatment 'replicate.' Growth rates of isolates pre- and post-phage treatment, and resistant and susceptible in vivo isolates were compared using 95% CIs that were estimated using a nonparametric bootstrap.

To explore genetic differences of in vitro isolates, we calculated several commonly used metrics using the frequency data of the 50 variants in known genes that differed significantly between our treatments: (1) the genetic distance from the ancestral population, calculated as the sum of the difference of the proportion of each SNP/indel in each population from the ancestral proportion; (2) the number of unique de novo SNPs/indels in each population; (3) alpha diversity, calculated using a modified version of the Hardy-Weinberg equilibrium, such that $\alpha = \sum \left( 1 - p_i^2 - q_i^2 \right)$, where i is the position of each SNP/indel, p is the proportion of the SNP/indel, and *q* is 1–p. This is equivalent to expected heterozygosity. Differences between these metrics were analyzed using linear models. To test whether consistent genetic differences occur within treatments, we performed nonmetric multidimensional scaling on the Euclidean distance matrix of SNPs/indels and their proportions in each population using '*metaMDS*' in the R package '*vegan.*' Nonmetric multidimensional scaling aims to collapse information from multiple dimensions (i.e., from different populations and multiple SNPs/indels per population) into just a few, allowing differences between samples to be visualized and interpreted. Permutational ANOVA tests were run using '*vegan::adonis,*' with Euclidean distance as the response term and treatment as the predictor variable with 9999 iterations. Changes in beta-diversity were examined using '*vegan::betadisper*' based on the same response and predictor variables as in the PERMANOVA.

For the in vivo sequencing, we looked for the presence of variants found to be significant in vitro in the in vivo isolates. To see if any of the genetic changes could explain differences in phenotypes of the isolates, we used nonmetric multidimensional scaling as described previously. Phenotypic traits of the isolates (virulence, growth rate, resistance profile, and biofilm production) were then mapped onto the resulting NMDS plot to see whether observed genetic changes correlated with any changes in phenotypes (*Figure 5—figure supplement 2*).

## Data cleaning

Before growth rate analyses, data points from the first 2.5 hr were removed as a sharp increase in OD was observed before lag phase independent of exponential growth (*Figure 3*; *Figure 3—figure supplement 1*).

In the in vitro data set, one replicate in the control and one in the susceptible treatments were found to be outliers that significantly impacted data analysis. In the susceptible treatment, the outlier (711) had a value 2.9× greater than the group mean (246.9) and this was 2.7× greater than the next

nearest value (266.1). Similarly, in the control group, the outlier (708.6) had a value 2.26× greater than the group mean (313.5) and this was 2.5× greater than the next nearest value (278). We subsequently determined the impact of these values individually and when together on model output by sequentially removing these from the data set (*Supplementary files 3 and 4*). Resistance was significant regardless of outlier removal (*Supplementary file 3*) as were comparisons between resistant and susceptible groups (*Supplementary file 4*). Although treatment became significant when the susceptible outlier was removed, all Tukey-HSD comparisons were nonsignificant. The only change detected in Tukey HSD comparisons was that between the single resistant and susceptible isolates which was found when the susceptible and both outliers were removed (*Supplementary file 4*).

## Acknowledgements

Genome sequencing was provided by MicrobesNG (http://www.microbesng.com). This work was supported in part by Grant MR/N0137941/1 awarded to MC for the GW4 BIOMED MRC DTP, awarded to the Universities of Bath, Bristol, Cardiff, and Exeter from the Medical Research Council (MRC)/UKRI. The work was also funded by a Royal Society Challenge Grant CH160068, a NERC standard Grant NE/S000771/1 awarded to AB, a Royal Higher Institute for Defence Grant HFM 19–12 awarded to MM, and a Grant from the Biotechnology and Biological Sciences Research Council (BBSRC) BB/T014342/1 awarded to VF.

## Additional information

### Funding

| Funder | Grant reference number | Author |
| --- | --- | --- |
| Medical Research Council | MR/N0137941/1 | Meaghan Castledine |
| Royal Society | CH160068 | Angus Buckling |
| Natural Environment Research Council | NE/S000771/1 | Angus Buckling |
| Royal Higher Institute for Defence | HFM 19-12 | Maya Merabishvili |
| Biotechnology and Biological Sciences Research Council | BB/T014342/1 | Ville-Petri Friman |

The funders had no role in study design, data collection and interpretation, or the decision to submit the work for publication.

### Author contributions

Meaghan Castledine, Conceptualization, Data curation, Formal analysis, Investigation, Methodology, Project administration, Validation, Visualization, Writing - original draft, Writing - review and editing; Daniel Padfield, Formal analysis, Investigation, Methodology, Project administration, Validation, Visualization, Writing - review and editing; Pawel Sierocinski, Conceptualization, Data curation, Investigation, Methodology, Project administration, Writing - review and editing; Jesica Soria Pascual, Data curation, Investigation, Methodology, Writing - review and editing; Adam Hughes, Lotta Mäkinen, Data curation, Investigation; Ville-Petri Friman, Conceptualization, Methodology, Validation, Visualization, Writing - review and editing; Jean-Paul Pirnay, Daniel de Vos, Conceptualization, Data curation, Methodology, Project administration, Resources, Supervision, Validation, Writing - review and editing; Maya Merabishvili, Conceptualization, Data curation, Resources, Writing - review and editing; Angus Buckling, Conceptualization, Formal analysis, Funding acquisition, Methodology, Project administration, Supervision, Validation, Writing - review and editing

### Author ORCIDs

Meaghan Castledine (iD) http://orcid.org/0000-0001-5752-2641
Daniel Padfield (iD) http://orcid.org/0000-0001-6799-9670

## Ethics

The decolonisation study protocol was approved by the Leading Ethics Committee of the "Université Catholique de Louvain" (Avis N°: B-403201111110). The study was performed in accordance with the ethical standards as laid down in the Declaration of Helsinki and as revised in 2013. The patients gave informed consent and their anonymity was preserved.

## Decision letter and Author response

Decision letter https://doi.org/10.7554/eLife.73679.sa1
Author response https://doi.org/10.7554/eLife.73679.sa2

## Additional files

### Supplementary files

• Supplementary file 1. Pairwise comparisons of resistance of in vivo clones from day two against phage from contemporary (day 2) and two future time points (days 4 and 7). P-values adjusted using the Tukey method of comparing a family of three estimates. "SE" = standard error.

• Supplementary file 2. Susceptible colonies picked from in vitro phage treatments for phenotypic assays and sequencing.

• Supplementary file 3. The effect of outlier removal on the significance of model terms (treatment and phage resistance). Outlier removed states in which treatment each outlier was identified and subsequently removed. Significant p-values in bold.

• Supplementary file 4. The effect of outlier removal on Tukey HSD contrasts comparing biofilm productivity between phage resistance groups. Significant p-values highlighted in bold.

• Transparent reporting form

### Data availability

All data and R script files can be found at the following GitHub repository. This repository outlines how the files may be used for analysis and Figure production. A full description of data file meanings and annotated R scripts is provided. https://github.com/mcastledine96/Parallel_evolution_phage_resistance_virulence_trade-offs_invivo_invitro, (copy archived at swh:1:rev:bf8ecd45bcb1f2293f022f-8392c0ad52ef6280c8). Raw sequencing files have been archived on the European Nucleotide Archive with the project accession number PRJEB47945 https://www.ebi.ac.uk/ena/browser/view/PRJEB47945?show=reads.

The following datasets were generated:

| Author(s) | Year | Dataset title | Dataset URL | Database and Identifier |
|---|---|---|---|---|
| Castledine M | 2021 | Parallel evolution of phage resistance - virulence trade - offs during in vitro and nasal *Pseudomonas aeruginosa* phage treatment | https://www.ebi.ac.uk/ena/browser/view/PRJEB47945?show=reads | ENA, PRJEB47945 |
| Castledine M | 2021 | Parallel evolution of *Pseudomonas aeruginosa* phage resistance and virulence loss in response to phage treatment in vivo and in vitro | https://github.com/mcastledine96/Parallel_evolution_phage_resistance_virulence_trade-offs_invivo_invitro | GitHub, GitHub |

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
