## [Editor Report]

With the increased interest in phage therapy to treat antibiotic resistant infections, there are questions about the ease at which bacteria evolve phage resistance. To examine this, Castledine et al. cultured a set of bacterial isolates from a patient pre- and during phage therapy and also experimentally evolved a mixture of the bacterial isolates from the patient in the absence or presence of phage in vitro. Overall, the authors observed similarities between the evolutionary outcomes (genomic and phenotypic) in the patient and in vitro.

---

## [Decision Letter]

**Decision letter after peer review:**

Thank you for submitting your article "Parallel evolution of phage resistance – virulence trade – offs during in vitro and nasal *Pseudomonas aeruginosa* phage treatment" for consideration by *eLife*. Your article has been reviewed by 2 peer reviewers, and the evaluation has been overseen by Gisela Storz serving as Reviewing and Senior Editor. The reviewers have opted to remain anonymous.

Essential revisions:

The reviewers thought this study was a strong candidate for *eLife* but suggest important edits and additional genomic analysis to improve the paper.

*Reviewer #1 (Recommendations for the authors):*

1. The title is difficult to parse. Perhaps change to: "Parallel evolutionary responses by *Pseudomonas aeruginosa* to phage therapy in vivo and in vitro lead to enhanced phage resistance and loss of virulence".

2. Line 79-80 this wording is slightly odd. Do you mean that the supply of genetic variation is determined by population size and gene flow?

3. Line 103 there is something up with the hyphens/dashes here. I think perhaps you are using the wrong symbol?

4. Line 131-132 this requires a citation e.g. Wright et al. 2018.

5. Figure 1. I find this figure difficult to understand; I can't understand why you use different formats for A versus B/C or why you have a line connecting the ancestral phages to the in vivo isolated phages in panel A? The simplest fix would be to change panel A to match the format of panel B/C.

6. Virulence assays: do these test virulence or the ability to establish acute infection i.e. did you sample bacteria from the waxmoth at the end to see if they did indeed establish infection?

7. Phenotype-genotype correlations: another caveat here is that you tested a limited number of phenotypes; adding motility, resource use etc. would have been interesting and are known to be affected by phage resistance.

8. Line 621 these references don't really support the statement about molecular evolution as neither did any sequencing? Perhaps Betts et al. Science 2018 would be better?

*Reviewer #2 (Recommendations for the authors):*

This work is a nice contribution to the literature on understanding bacteria-phage interactions in different conditions. Below I have included some questions that remained for me after reading the manuscript, and specific suggestions for improving clarity.

1. In general, I often found it hard to follow where isolates were coming from for the different assays. An overview schematic/diagram figure might help clarify.

2. I thought it was particularly interesting that evolutionary similarities were found using LB as the in vitro media (and not a media type meant to replicate the in vivo environment). Could be interesting to highlight or at least mention in the discussion.

3. Line 124: "The phage cocktail included a third phage that specifically infects *S. aureus* bacterium but was not detected in this patient." Was S. aureus or the third phage not detected in the patient? The language is unclear to me.

4. Line 149: Could the authors comment on why the "repeated" treatment was done using added phage every two transfers, with the transfers occurring two days apart? I am curious why this was chosen rather than applying phage daily as was done in vivo.

5. Line 162: An MOI of 0.5 was used in vitro – is there any idea/estimate of whether this was similar to what was applied in vivo?

6. Lines 184-186: I am curious about what the results would look like if the bacterial resistance/susceptibility data were not condensed to this binary metric (given that the authors have access to this information!). Is there any relationship between # of phages an isolate is resistant to and the other traits measured in this study?

7. Line 191: Could the authors comment more on the relevance of this model for virulence in humans (possibly in the discussion)?

8. Line 368: "In subsequent tests of phage resistance to phage isolates isolated in vivo" I found this wording unclear.

9. Line 374: Is "7" meant to be "4"?

10. Line 383 Figure 1: The text is written mostly in terms of bacterial resistance, but the y-axis in this figure is plotting susceptibility. I think it would be easier to follow if the y-axis is proportion resistant.

11. Line 403: "there was a significantly higher proportion of phage resistant bacteria in the treatment where phages were added once compared to when phages were added repeatedly" I think as written this is a little misleading – I am assuming (although not sure) that this is comparing proportion of T3 bacterial resistance to the T3 phages from the different treatments (which are different sets of phages). When comparing resistance to the same phages (e.g. the ancestral phages as shown in Figure 1b and 1c), bacteria from the "repeated" treatment have similar or even possibly higher levels of resistance. Note this was also written similarly in lines 650-651.

12. Line 453 Figure 2: Why is there no legend below panel a?

13. Line 501 Figure 3d: Given that the significant result here was that phage resistant isolates had increased biofilm production compared to phage sensitive isolates, it would be nice to see the boxplots broken down by resistance/sensitivity.

14. Line 602-603: "That resistance emerged predominantly by de novo mutations and not selection on existing variation" how exactly was this concluded?

15. Various possible typos/errors: line 151 ("assessed"), lines 247-248 ("called identified"), line 265 ("were occurred"), lines 267-268 ("DNA extraction and whole genome sequencing on each isolate individually"), lines 378-379 ("apart from to"), line 455 ("bacterial").

---

## [Author Response]

Reviewer #1 (Recommendations for the authors):1. The title is difficult to parse. Perhaps change to: "Parallel evolutionary responses by *Pseudomonas aeruginosa* to phage therapy in vivo and in vitro lead to enhanced phage resistance and loss of virulence".

We agree this title could have been more accessible and have changed this to, “Parallel evolution of *Pseudomonas aeruginosa* phage resistance and virulence loss in response to phage treatment in vivo and in vitro”.

2. Line 79-80 this wording is slightly odd. Do you mean that the supply of genetic variation is determined by population size and gene flow?

Here, we intended to convey that mutation supply is dependent on resources as this facilitates a rapid growth rate and, you rightly state, supports a large population size – as such there is a greater number of random mutations. This sentence has been clarified on line 87:

“In order to evolve phage resistance, bacteria must have sufficient mutation supply, which will primarily be determined by population size and gene flow (Gandon et al., 2008; Hampton et al., 2020; Morgan et al., 2010, 2005; Pal et al., 2007).”

3. Line 103 there is something up with the hyphens/dashes here. I think perhaps you are using the wrong symbol?

Thank you for spotting this, these symbols have been corrected.

4. Line 131-132 this requires a citation e.g. Wright et al. 2018.

This citation has been added.

5. Figure 1. I find this figure difficult to understand; I can't understand why you use different formats for A versus B/C or why you have a line connecting the ancestral phages to the in vivo isolated phages in panel A? The simplest fix would be to change panel A to match the format of panel B/C.

As in vivo samples follow individual bacterial clones, observed data points in panel A would have a value of 0 or 1. In contrast, for our in vitro work we were able to sample multiple bacterial clones from individual treatment replicates to give an average resistance level for each replicate. We aimed to show as much consistency as possible between the panels by showing the mean levels of resistance for each bacterial population to phage. Lines do not connect ancestral to other phage time’s in panels B and C as ancestral levels of resistance were measured using a separate population of bacterial clones isolated at the end of the experiment.

6. Virulence assays: do these test virulence or the ability to establish acute infection? i.e. did you sample bacteria from the waxmoth at the end to see if they did indeed establish infection?

We did not sample bacteria at the end of infection, but our measure of virulence is consistent with classic definition, *i.e.*, increased host mortality resulting from parasite infection (May and Anderson 1990). We have clarified this in the methods (lines 593-594).

7. Phenotype-genotype correlations: another caveat here is that you tested a limited number of phenotypes; adding motility, resource use etc. would have been interesting and are known to be affected by phage resistance

We have now highlighted these as other potential phenotypes that could have also been measured (lines 436-439). We have also stated this was beyond the scope of this work owing to the volume of data presented and we do not think these assays would not change the conclusions significantly.

8. Line 621 these references don't really support the statement about molecular evolution as neither did any sequencing? Perhaps Betts et al. Science 2018 would be better?

Although Pal et al. (2007) and Morgan et al. (2010) did not do sequencing, they showed phage select for mutator strains of bacteria which are shown to accelerate molecular evolution. We have added Betts et al. (2018) as further support for this statement.

Reviewer #2 (Recommendations for the authors):This work is a nice contribution to the literature on understanding bacteria-phage interactions in different conditions. Below I have included some questions that remained for me after reading the manuscript, and specific suggestions for improving clarity.1. In general, I often found it hard to follow where isolates were coming from for the different assays. An overview schematic/diagram figure might help clarify.

We agree this is challenging to follow and have included a figure (Figure 6) to clarify how colonies evolved in vitro were isolated for each assay.

2. I thought it was particularly interesting that evolutionary similarities were found using LB as the in vitro media (and not a media type meant to replicate the in vivo environment). Could be interesting to highlight or at least mention in the discussion.

We have now highlighted this on lines 382-384.

3. Line 124: "The phage cocktail included a third phage that specifically infects *S. aureus* bacterium but was not detected in this patient." Was S. aureus or the third phage not detected in the patient? The language is unclear to me.

Apologies for the confusion. Here, we meant to indicate the bacteria (*S. aureus*) and this has been edited for clarity 509-510.

4. Line 149: Could the authors comment on why the "repeated" treatment was done using added phage every two transfers, with the transfers occurring two days apart? I am curious why this was chosen rather than applying phage daily as was done in vivo.

We struggled with this decision. Our rationale was that phage would be rapidly lost from the nasal cavity (immunity, breathing, dispersal of mucous), while the in vitro environment is a closed system. We were concerned about obscuring any phage evolution in the latter by continually flooding the system with large numbers of ancestral phages. As a result, we chose to apply phage every second day in vivo (now clarified on lines 536-537).

5. Line 162: An MOI of 0.5 was used in vitro – is there any idea/estimate of whether this was similar to what was applied in vivo?

It is unfortunately too difficult to accurately estimate *P. aeruginosa* densities from a single swab or even the actual amount of phage that successfully settled in the respiratory tract from a nebuliser. However, the aim was to establish a relatively high MOI in vivo with a phage density of 10^8^ pfu/mL in the nebuliser solution.

*6. Lines 184-186: I am curious about what the results would look like if the bacterial resistance/susceptibility data were not condensed to this binary metric (given that the authors have access to this information!). Is there any relationship between # of phages an isolate is resistant to and the other traits measured in this study?*

Unfortunately, while this would be interesting to investigate, we do not have access to this information – bacterial resistance was measured as a binary variable (using spotting assays) against phage populations at each time point.

7. Line 191: Could the authors comment more on the relevance of this model for virulence in humans (possibly in the discussion)?

Waxmoth larvae lack the adaptive immune system of vertebrates, including humans, but it is argued that similarities in innate immune systems allow to be a comparable model for acute infections. As the infection in our study was a nasal colonisation, lacking adaptive immune system involvement, we believe this is an appropriate model for virulence. We have highlighted this on lines 591-593.

8. Line 368: "In subsequent tests of phage resistance to phage isolates isolated in vivo" I found this wording unclear.

Apologies, we have clarified this wording (lines 131-132).

9. Line 374: Is "7" meant to be "4"?

This has been corrected.

10. Line 383 Figure 1: The text is written mostly in terms of bacterial resistance, but the y-axis in this figure is plotting susceptibility. I think it would be easier to follow if the y-axis is proportion resistant.

We agree this would match with the text more closely and have adjusted the figure to proportion resistant.

11. Line 403: "there was a significantly higher proportion of phage resistant bacteria in the treatment where phages were added once compared to when phages were added repeatedly" I think as written this is a little misleading – I am assuming (although not sure) that this is comparing proportion of T3 bacterial resistance to the T3 phages from the different treatments (which are different sets of phages). When comparing resistance to the same phages (e.g. the ancestral phages as shown in Figure 1b and 1c), bacteria from the "repeated" treatment have similar or even possibly higher levels of resistance. Note this was also written similarly in lines 650-651.

We agree this could be confusing or misleading. We have clarified on lines 167 and 448 that this difference is where resistance is measured to contemporary phage populations. As the contemporary phage population is the active infecting population of phage, we would argue this is the most informative measure of phage resistance to report at the end of the experiment.

12. Line 453 Figure 2: Why is there no legend below panel a?

A legend has now been added.

13. Line 501 Figure 3d: Given that the significant result here was that phage resistant isolates had increased biofilm production compared to phage sensitive isolates, it would be nice to see the boxplots broken down by resistance/sensitivity.

We respectfully disagree but can change this if required. We think that it is more appropriate to separate the boxplots by treatment, both because this is what was manipulated and for consistency with other figures. The different symbols for susceptible and resistant isolates make the difference in biofilm production clear.

14. Line 602-603: "That resistance emerged predominantly by de novo mutations and not selection on existing variation" how exactly was this concluded?

There were many more de novo mutations in the in vitro phage treatments than the no phage control had 5-10 times more de novo mutations than control populations, suggesting de novo mutations were an important determinate of phage resistance. We have clarified this on lines 352-354.

“In total, 31 of 50 genetic changes (62%) in vitro were found in vivo. Of these, 77% were de novo mutations in vitro, indicating that de novo mutations were also important for phage resistance in vivo.”

15. Various possible typos/errors: line 151 ("assessed"), lines 247-248 ("called identified"), line 265 ("were occurred"), lines 267-268 ("DNA extraction and whole genome sequencing on each isolate individually"), lines 378-379 ("apart from to"), line 455 ("bacterial")

Thank you for spotting these. These have now all been corrected.